# Fecal Carriage of Extended-Spectrum β-Lactamase-/AmpC-Producing *Escherichia coli* in Pet and Stray Cats

**DOI:** 10.3390/antibiotics12081249

**Published:** 2023-07-29

**Authors:** Gabriele Ratti, Alessia Facchin, Angelica Stranieri, Alessia Giordano, Saverio Paltrinieri, Paola Scarpa, Deborah Maragno, Alessia Gazzonis, Martina Penati, Camilla Luzzago, Paola Dall’Ara, Stefania Lauzi

**Affiliations:** 1Department of Veterinary Medicine and Animal Sciences (DIVAS), University of Milan, 26900 Lodi, Italy; gabriele.ratti@unimi.it (G.R.); alessia.facchin@unimi.it (A.F.); angelica.stranieri@unimi.it (A.S.); alessia.giordano@unimi.it (A.G.); saverio.paltrinieri@unimi.it (S.P.); paola.scarpa@unimi.it (P.S.); deborah.maragno@studenti.unimi.it (D.M.); alessia.gazzonis@unimi.it (A.G.); martina.penati@unimi.it (M.P.); camilla.luzzago@unimi.it (C.L.); stefania.lauzi@unimi.it (S.L.); 2Laboratory of Animal Infectious Diseases (MiLab), University of Milan, 26900 Lodi, Italy

**Keywords:** cats, antimicrobial resistance, extended-spectrum β-lactamase, AmpC, *E. coli*, resistance genes

## Abstract

Dogs have been reported as potential carriers of antimicrobial-resistant bacteria, but the role of cats has been poorly studied. The aim of this study was to investigate the presence and the risk factors associated with the fecal carriage of extended-spectrum β-lactamase and AmpC (ESBL/AmpC)-producing *Escherichia coli* (*E. coli*) in pet and stray cats. Fecal samples were collected between 2020 and 2022 from healthy and unhealthy cats and screened for ESBL/AmpC-producing *E. coli* using selective media. The presence of ESBL/AmpC-producing *E. coli* was confirmed by phenotypic and molecular methods. The evaluation of minimum inhibitory concentrations (MICs) was performed on positive isolates. Host and hospitalization data were analyzed to identify risk factors. A total of 97 cats’ samples were collected, and ESBL/AmpC-producing *E. coli* were detected in 6/97 (6.2%), supported by the detection of *bla*_CTX-M_ (100%), *bla*_TEM_ (83.3%), and *bla*_SHV_ (16.7%) genes and the overexpression of chromosomal *ampC* (1%). All *E. coli* isolates were categorized as multidrug-resistant. Unhealthy status and previous antibiotic therapy were significantly associated with ESBL/AmpC-producing *E. coli* fecal carriage. Our results suggest that cats may be carriers of ESBL/AmpC-producing *E. coli*, highlighting the need for antimicrobial stewardship in veterinary medicine and an antimicrobial-resistance surveillance program focusing on companion animals, including stray cats.

## 1. Introduction

Antimicrobial resistance (AMR) has become one of the major threats to public health [1]. Multisectoral surveillance systems focusing on resistance to antibiotics are key points to define and evaluate the effectiveness of measures against AMR. Due to their rapid emergence over the last few years, concerns have been raised about the spread of extended-spectrum beta-lactamase (ESBL)- and AmpC-producing *Escherichia coli* (*E. coli*) in both humans and animals [2,3]. The importance of ESBL/AmpC-producing *E. coli* strains has been emphasized by their association with increasing treatment failure, hospitalization, and mortality in humans and animals [4]. The need for a One Health approach has been highlighted [5], and ESBL/AmpC-producing *E. coli* have been proposed as commensal indicator microorganisms for AMR surveillance systems [5,6].

Most of the studies and national surveillance programs related to AMR have focused on food-producing animals [7,8,9]. However, the close contact and sharing of environments between humans and companion animals as well as the wide spread of dogs and cats in European households has raised concerns about the role of companion animals in AMR dissemination and the potential risk of the transmission of resistant bacteria to humans or vice versa [2,10]. The presence of ESBL-producing *E. coli* has been reported among dogs worldwide, with different prevalence levels likely reflecting differences in the diagnostic methods used; the levels of antibiotic use among veterinarians and owners; and other factors influencing bacterial transmission, including socioeconomic and behavioral components [11]. Similarly to humans and food-producing animals, the most frequent genes associated with ESBL resistance in dogs encode for CTX-M enzymes, followed by TEM and SHV [12,13]. Moreover, the presence of AmpC-producing *E. coli* in dogs, usually carrying the *bla*_CMY-2_ gene, has been reported with a lower prevalence compared to ESBL-producing *E. coli* [3,13,14].

Despite being the most popular companion animals in Europe, few studies have investigated the presence of ESBL/AmpC-producing *E. coli* in cats [11,15]. The presence of ESBL/AmpC-producing *E. coli* in cats has been reported in clinical samples from diseased cats (with a variety of common clinical conditions, including gastrointestinal disease, upper respiratory tract disease, otitis, conjunctivitis, stomatitis, skin abscess, and urinary tract infections) and in fecal samples from healthy cats [3,14,16,17]. In addition to owned cats, stray cats have also been reported as reservoirs of AMR *E. coli* [18]. The role of host factors associated with the spread of ESBL/AmpC-producing *E. coli* in cats still needs to be elucidated [2,19]. Given the importance of cats as a potential ESBL/AmpC-producing *E. coli* reservoir because of their wide diffusion as pet animals and their close contact and sharing of environments with humans, the aims of this study were to estimate the presence of ESBL/AmpC-producing *E. coli* fecal carriage in cats, characterize the antimicrobial-resistance phenotypes and genotypes of the isolates, and identify risk factors associated with the fecal carriage of ESBL/AmpC-producing *E. coli*. 

## 2. Results

In total, 97 fecal samples from cats were included in this study. The characteristics of the analyzed cats are summarized in Table 1. Statistical analysis showed that the unhealthy status of animals (OR = 5.91; 95% CI: 1.01–34.44; *p* = 0.049) and previous antibiotic therapy (OR = 6.67; 95% CI: 1.14–38.99; *p* = 0.037) were significantly associated with the fecal carriage of ESBL/AmpC-producing *E. coli*, whereas the other variables considered in the analysis were not risk factors (Table 1). The two positive stray cats detected in this study originated from two different feline colonies.

The characteristics of the unhealthy cats compared to healthy cats analyzed in this study are given in Appendix A.

Overall, the presence of ESBL/AmpC-producing *E. coli* was detected in 6/97 (6.2%; 95% confidence interval: 1.4–11%) cats, and the characteristics of the positive cats are reported in Table 1. More precisely, ESBL-producing *E. coli* was only observed in five (83.3%) isolates, whereas ESBL- and AmpC-producing *E. coli* were observed in one (16.7%) strain only. The results of the phenotypic and genetic characterization of the *E. coli* isolates are summarized in Table 2. Among the *bla*_CTX-M_-positive isolates, all *E. coli* isolates carried the *bla*_CTX-M-1_ group gene.

The analysis of the promoter/attenuator region of the AmpC-producing *E. coli* detected in this study showed the presence of a −32 T > A mutation in the −35 promoter box and a −28 G > A mutation in the spacer region (Appendix A), previously associated with the upregulation of AmpC production [20].

Based on MIC results (Table 3), the highest level of resistance was observed for β-lactam, with all isolates resistant to ampicillin, cefazolin, cephalexin, cefovecin, and cefpodoxime, and for doxycycline. Resistance to fluoroquinolones (16.7%); phenicol (16.7%); β-lactam in combination with the β-lactamase inhibitor agent (16.7% and 50% for amoxicillin/clavulanic acid 2:1 and piperacillin/tazobactam content 4, respectively); tetracycline (83.3%); and carbapenems (16.7%) was less frequent, with at least one isolate resistant to all the agents in each category. All isolates were susceptible to amikacin, accounting for the lowest resistance rate. All ESBL/AmpC-producing *E. coli* isolates detected in this study were MDR.

## 3. Discussion

Cats are the most common companion animals worldwide, but only a few studies have investigated their role as potential carriers of AMR bacteria [7,11,21]. In the present study, fecal samples were analyzed from stray and pet cats to estimate the presence of ESBL/AmpC-producing *E. coli* fecal carriage in cats from Italy and the associated risk factors. The overall presence of ESBL/AmpC-producing *E. coli* in 6.2% of cats detected in this study was consistent with previous reports that have investigated the fecal carriage of ESBL-producing *E. coli* in healthy and sick cats from Portugal [3] and in sheltered cats [22]. The recent meta-analysis performed by Salgado-Caxito and colleagues [11] showing that the global estimated prevalence of ESBL-producing *E. coli* in cats is 5.04% (95% CI: 2.42–10.22%) also confirms our results. However, other studies have detected a prevalence of up to 20% for ESBL-producing *E. coli* among cat fecal samples [23,24], probably due to the different backgrounds of the tested cats or the different diagnostic methods used. Indeed, a comparison between studies may be challenging due to the lack of standardized diagnostic methods for AMR surveillance in pets [11,13]. Therefore, according to standardized protocols for food-producing animals [6], the implementation of a specific standardized diagnostics methodology for AMR surveillance programs in companion animals may be suggested. 

The presence of the ESBL genotype supported by the detection of *bla*_CTX-M_ as the main resistance gene in all isolates in this study, followed by *bla*_TEM_ and *bla*_SHV,_ confirmed previous studies reporting these as the most common genes in ESBL-producing bacteria from both humans and animals [25,26], though it should be noted that we did not determine the TEM and SHV groups of the isolates, which do not always encode for ESBL [27]. Our results showing that the majority of *E. coli* isolates carried more than one resistance gene confirmed the genetic diversity of ESBL-producing *E. coli* in companion animals [11,13]. Moreover, the high detection rate of *bla*_CTX-M-1_ group genes was expected, as this group is the most common worldwide [28].

The presence of mutations of the AmpC-producing *E. coli* isolates detected in this study associated with the upregulation of AmpC production [20] was consistent with previous reports highlighting the presence of *campC* resistance in companion animals [29,30]. However, it must be noted that AmpC hyperproduction is less frequently investigated compared to plasmid-mediated AmpC resistance, since the latter mechanism is capable of being horizontally transferred to other bacteria, posing a greater threat to AMR control [20,31]. Indeed, the absence of *pampC* among the AmpC-producing *E. coli* detected in this study could have been due to the limited sample size and the lower presence of pAmpC-producing *E. coli* compared to ESBL-producing *E. coli* in companion animals [3,13,14,17,32]. Therefore, further studies with a wider sample size are necessary to assess the role of cats in the dissemination of pAmpC-producing *E. coli.* Moreover, our results confirmed that commercially available kits for AmpC detection failed to differentiate between pAmpC and cAmpC production in *E. coli,* as previously reported [31].

Interestingly, one isolate detected in this study that carried *bla*_CTX-M_ and *bla*_TEM_ genes was phenotypically negative for ESBL production. The ESBL phenotype could have been masked by the presence of AmpC β-lactamases and/or carbapenemases [33]. However, the presence of AmpC β-lactamases was unlikely, as the isolate was negative for the AmpC phenotype. Interestingly, this isolate was the only one resistant to imipenem, with an MIC value ≥16 µg/mL. A high imipenem MIC value has been used as a screening method for carbapenemase production in *Enterobacteriaceae* [34,35], as isolates with an MIC value of 2 µg/mL or above are likely to be carbapenemase producers [36,37]. Unfortunately, the aim of our study did not include the detection of carbapenemase-producing bacteria. In this respect, further studies are needed to clarify the epidemiological role of cats in the carriage of carbapenemase-producing *E. coli*, as the presence of carbapenemase-producing *Enterobacteriaceae* in companion animals has been rarely investigated, despite recent studies having shown infection or colonization by carbapenemase-producing Enterobacterales in companion animals [21,38,39].

Regarding *E. coli* phylogroups, the detection of B2 as the most frequent phylogenetic group was consistent with previous reports showing a high presence of phylogroup B2 among the AMR *E. coli* in companion animals and humans but not in food-producing animals, suggesting the need for further investigations to understand the zoonotic potential of these isolates [17,40,41,42,43,44]. Moreover, given that *E. coli* belonging to phylogroups B2 and F has been commonly associated with the presence of virulence factors responsible for extra-intestinal infections [40,44], further investigations are suggested to assess the virulotyping of ESBL/AmpC-producing *E. coli* in cats.

The high presence of MDR *E. coli* was expected, as ESBL-producing *E. coli* usually not only exhibit resistance to β-lactam, but frequently carry other resistance genes conferring resistance to other antimicrobial drugs, leading to MDR [45]. Moreover, the high detection rate of MDR *E. coli* highlighted the need for antimicrobial stewardship in veterinary practice to also reduce the emergence and spread of MDR bacteria in cats.

The unhealthy status of cats as a risk factor associated with the fecal carriage of ESBL/AmpC-producing *E. coli* observed in our study was likely influenced by the fact that a higher percentage of sick animals were treated with antibiotics compared to healthy cats. This suggests that antibiotic treatment alone may have increased the risk of the fecal carriage of AMR organisms in this study, as it is a well-recognized risk factor for the acquisition of drug-resistant bacteria in both humans and animals [2,46,47]. It was not possible to define if specific antibiotic classes were associated with ESBL/AmpC-producing *E. coli* due to the low number of cats treated with antibiotics in this study. The absence of an association between hospitalization and the fecal carriage of ESBL/AmpC-producing *E. coli* could have been due to the low number of hospitalized cats included in the present study or to the differences in management and hygiene protocols among hospitals [19], since hospitalization has been associated with the carriage of drug-resistant bacteria in companion animals [19,46,47]. In this respect, further studies focusing on hospitalized cats are needed to evaluate the contribution of veterinary hospitals in the spread of drug-resistant bacteria. 

The presence of ESBL/AmpC-producing *E. coli* among both stray and owned cats is of interest and confirms a recent report detecting AMR in stray cats [18]. Given the unknown backgrounds of the stray cats in this study, we could not conclude if the AMR acquisition was associated with environmental sources, antimicrobial administration, the presence of subclinical disease (cats were healthy at physical examination), or other factors. Moreover, the apparent non-statistically significant higher ESBL/AmpC-producing *E. coli* carriage in owned cats compared to stray cats may have been influenced by the high percentage of unhealthy animals among the owned cats, likely requiring antimicrobial therapy. Indeed, although this group of cats was raised in an enclosed environment, with a clean water and food supply and a good level of care from their owners, more than half of the owned cats were unhealthy, whereas all stray cats were apparently healthy and were brought to the VTH only for neutering.

Further investigations are needed to understand the potential route of transmission of AMR in stray cat populations as well as to clarify the role of stray cats, as previously reported for stray dogs in South America [14,48].

This study had some limitations, starting from the low number of analyzed cats, which was not sufficient for population estimation. Further studies with a greater sample size and an ad hoc sampling strategy are needed to assess the epidemiological role of stray and owned cats in the spread of ESBL/AmpC-producing *E. coli*. The whole-genome sequencing of the *E. coli* isolates detected in this study is suggested in order to define virulence factors and the presence of other resistance genes that were not investigated in this study. Moreover, it would be interesting to analyze more than one AMR isolate from each sample to investigate if the genetic variability of *E. coli* isolates can occur within the same fecal sample.

## 4. Materials and Methods

### 4.1. Sample Collection

Fecal samples were collected from both owned and stray cats admitted to the Veterinary Teaching Hospital (VTH) of Lodi, University of Milan, Italy from 2020 to 2022. Fecal samples from owned cats were collected from leftover material submitted to the VTH laboratory for diagnostic purposes or collected for research purposes in collaboration with local veterinarians. Stray cats were admitted to undergo neutering programs for the demographic control of stray population. Samples were stored at 4 °C until arrival at the laboratory (within 24 h from collection). The study was approved by the Institutional Animal Care and Use Committee and the Institutional Ethical Committee (approval no. OPBA_40_2020). Furthermore, residual fecal samples from cats collected for diagnostic purposes at the VTH with the informed consent of the owners were used for this study without any additional formal request for authorization, according to the decision of the Ethical Committee of the University of Milan (EC decision 29 October 2012, renewed with protocol no. 02-2016).

### 4.2. Identification of ESBL/AmpC-Producing E. coli

Samples were screened for the presence of ESBL/AmpC-producing *E. coli* according to the standard protocol of the DTU National Food Institute, the reference laboratory for antimicrobial resistance in Europe, with minor modifications [49]. One gram of each fecal sample was pre-enriched in buffered peptone water (BPW) and incubated at 37 ± 1 °C for 18–22 h. For ESBL/AmpC-producing *Enterobacteriaceae*, a loopful of BPW was inoculated on MacConkey agar supplemented with 1 mg/L cefotaxime and incubated overnight at 37 ± 1 °C. Up to four bacterial colonies from positive growths of each sample were submitted to species identification regardless of colony morphology (lactose +/−). Species identification was accomplished in duplicate via matrix-assisted laser desorption/ionization time-of-flight mass spectrometry (MALDI-TOF MS) (MBT Microflex ^®^ LT/SH MALDI-TOF mass spectrometer, Bruker Daltonik, GmbH, Bremen, Germany) using the direct transfer method [50]. Following species identification, all four colonies/samples recovered from MacConkey agar supplemented with 1 mg/L cefotaxime were stored in brain heart infusion (BHI) broth with 15% glycerol at −80 °C.

One confirmed *E. coli* isolate for each sample was further thawed and used for ESBL and AmpC phenotypes confirmation using a combination disk test (CDT) and AmpC detection set D69C (MAST Group Ltd., Bootle, UK), respectively, according to EUCAST guidelines [33]. Briefly, a pure fresh culture of the tested isolate was suspended in physiological saline to obtain a 0.5 McFarland standard density equivalent suspension. The suspension was spread uniformly across the surface of a Mueller Hinton agar plate. Disks used for ESBL phenotyping contained ceftazidime or cefotaxime (30 μg) with and without clavulanic acid (10 μg). Disks used for AmpC phenotyping were disks A, B, and C. Disks were placed on the inoculated medium and incubated at 35 ± 1 °C for 18 ± 2 h. Results were interpreted following the manufacture’s instructions. For ESBL phenotyping, the isolate was considered positive in the presence of a ≥5 mm increase in the inhibition zone diameter in the disk containing ceftazidime or cefotaxime and clavulanic acid compared to the one without clavulanic acid. For AmpC phenotyping, the isolate was considered positive if the difference in the inhibition zone diameter between disks C and A and between disks C and B was ≥5 mm.

Isolates were also subjected to PCR analysis and the evaluation of minimum inhibitory concentrations (MICs).

### 4.3. PCR Analysis for Resistance Genes and E. coli Phylogroup

Single thawed *E. coli* colonies recovered from MacConkey agar supplemented with 1 mg/L cefotaxime were resuspended in 100 µL of sterile distilled water, and DNA was extracted by the boiling method at 95 °C for 10 min and subjected to PCR analysis to determine ESBL/AmpC genes and *E. coli* phylogroup.

For the detection of ESBL-producing *E. coli*, a multiplex PCR targeting *bla*_CTX-M_, *bla*_TEM_, and *bla*_SHV_ was performed on the DNA of all isolates, as previously reported [51]. Isolates positive for *bla*_CTX-M_ genes were further analyzed with a specific PCR for the *bla*_CTX-M-1_ group, *bla*_CTX-M-2_ group, *bla*_CTX-M-8_ group, *bla*_CTX-M-9_ group, and *bla*_CTX-M-25_ group, as previously reported [52,53,54,55,56]. Since the ESBL phenotype could be masked by the presence of AmpC β-lactamases and/or carbapenemases, cats were considered positive for the presence of ESBL-producing *E. coli* in the presence of at least one ESBL-encoding gene, regardless of phenotypic confirmation, as previously reported [33,57].

The DNA of *E. coli* isolates showing the AmpC phenotype were subjected to PCR for the detection of major plasmid-mediated *ampC* β-lactamase (*pampC*)-encoding genes and chromosomal-mediated *ampC* β-lactamases (*campC*). The presence of *pampC* was determined using a multiplex PCR, as previously reported [58]. The presence of *campC* hyperproduction was investigated by the amplification of a 271 bp fragment of the promoter/attenuator region, as previously reported [59]. Amplicons were purified and Sanger sequenced by a commercial sequencing facility (Microsynth Seqlab, Göttingen, Germany). The sequences were aligned against the promoter/attenuator region of the *campC* gene of *E. coli* strain ATCC 25922 using Clustal X in BioEdit software v.7.0. Strains were labeled *campC* hyperproducers when promoter mutations were found, according to previous reports [20,60]. *E. coli* isolates showing the AmpC phenotype supported by the overexpression of *campC* or by the presence of *pampC* genes were considered as AmpC-producing *E. coli.*

The *E. coli* phylogenetic group was determined following previously published protocols [44].

The different PCR assays were performed using positive control strains, and a blank control (DNAse-free water sample) was also included in all the PCR reactions. The primers used in this study are shown in Appendix A.

### 4.4. Antimicrobial Susceptibility Testing

MICs were determined on single ESBL- and/or AmpC-producing *E. coli* isolates by the broth microdilution method using a commercially available plate (COMPGN1F Sensititre plates, Thermo Fisher Scientific^®^, Waltham, MA, USA). The list of antimicrobials used, the related cut-off values, and the MIC (µg/mL) range are reported in Appendix A: *E. coli* ATCC 25922 was used as the control strain for susceptibility testing. Results were defined manually using a Sensititre Manual Viewbox (Sensititre™, Thermo Fisher Scientific^®^, Waltham, MA, USA). The MIC results were interpreted according to Clinical and Laboratory Standard Institute breakpoints CLSI VET01S ED6:2023 [61] following the manufacturer’s instructions. Multidrug resistance (MDR) was defined as non-susceptibility to at least one antimicrobial agent in three or more antimicrobial categories, as previously reported [62].

### 4.5. Data Analysis

For each animal, information regarding sex; age; type of ownership; and clinical history (clinical status, history of hospitalization, and previous antibiotic treatment performed within three months) were collected, as well as the results of biochemical analyses on blood and serum samples, if performed as part of the VTH diagnostic procedures. The stray cats belonged to 11 cat colonies from Lodi province, and the geographic location of the feline colony of origin was recorded. For the age variable, two categories were considered: <2 years old and ≥2 years old, as previously reported for the identification of age as a risk factor for infectious diseases [63]. The clinical status variable was classified into two categories: healthy and unhealthy, according to the presence/absence of a clinical manifestation and the results of routine laboratory analyses, if available. The clinical status of unhealthy cats was further classified, according to the main clinical presentation on admission, into gastrointestinal, respiratory, urogenital, dermatological, neurologic, traumatic, or systemic. Previous antibiotic treatment was further classified according to the antibiotic class used to treat the cats.

Pearson’s chi-square test and Fisher’s exact probability test were used to evaluate the differences between the proportions of ESBL/AmpC-producing *E. coli*-positive cats and sex, age, type of ownership, clinical status, hospitalization, and previous antibiotic therapy. Statistical comparisons were carried out using Epitools (https://epitools.ausvet.com.au/ (accessed on: 16 February 2023)), taking *p* < 0.05 as significant.

## 5. Conclusions

In this study, the fecal carriage of ESBL/AmpC-producing *E. coli* was detected in both owned and stray cats from Italy, with an estimated positivity of 6.2%, confirming that cats previously treated with antibiotics are at a higher risk of AMR carriage. These results highlight the need for antimicrobial stewardship in veterinary medicine and an AMR surveillance program focusing on companion animals, including stray cats.

## Figures and Tables

**Table 1 antibiotics-12-01249-t001:** Characteristics of cats analyzed in this study.

Population Characteristics		No. (%)	No. ESBL/AmpC-Positive (%)	*p* Value ^a^
Sex	Male	54 (55.7)	5 (9.3)	0.22
	Female	43 (44.3)	1 (2.3)	
Age ^b^	<2 years	55 (57.9)	3 (5.5)	1
	≥2 years	40 (42.1)	3 (7.5)	
Type of ownership	Stray	50 (51.5)	2 (4.0)	0.43
	Owned	47 (48.5)	4 (8.5)	
Clinical status	Healthy	70 (72.2)	2 (2.9)	**0.049**
	Unhealthy	27 (27.8)	4 (14.8)	
Clinical syndrome at admission	Gastrointestinal	12 (44.4)	2 (16.7)	nd
	Respiratory	8 (29.6)	1 (12.5)	
	Urogenital	2 (7.4)	0 (0)	
	Systemic	2(7.4)	1 (50)	
	Dermatological	1 (3.7)	0 (0)	
	Neurologic	1 (3.7)	0 (0)	
	Traumatic	1 (3.7)	0 (0)	
Hospitalization	Yes	10 (10.3)	2 (20.0)	0.11
	No	87 (89.7)	4 (4.6)	
Previous antibiotic therapy	Yes	25 (25.8)	4 (16)	**0.037**
	No	72 (74.2)	2 (2.8)	
Antibiotic class used in treated cats ^c^	Fluoroquinolones	12 (48.0)	0 (0)	nd
	β-Lactams and β-Lactamase inhibitors	10 (40.0)	1 (10.0)	
	Cephalosporins	7 (28.0)	2 (28.6)	
	Macrolide-nitroimidazole	2 (8.0)	1 (50.0)	
	Tetracyclines	2 (8.0)	0 (0)	

^a^ Numbers in bold indicate *p* < 0.05. ^b^ Age was unknown for two cats. ^c^ Cats treated with single and combination therapy were included. nd = not determined. Statistical analysis was not performed due to the low numbers of cats included in the different groups.

**Table 2 antibiotics-12-01249-t002:** Phenotypic and genetic characterization of ESBL/AmpC-producing *E. coli* isolates.

Isolate ID	Animal Characteristics						Phenotype	Genetic Determinants of Resistance	Phylogenetic Group	Resistance Pattern
	Sex	Age	Ownership	Clinical Status	Hospitalization	Previous Antibiotic Therapy (Antibiotic)				
46/1	Female	6 months	Stray	Healthy	No	No	ESBL/AmpC	*bla*_CTX-M-1_ group, *campC* hyperproducer	B2	AMP, FAZ, FOV, POD, LEX, AUG2, DOX, TET
51/1	Male	3 years	Stray	Healthy	No	No	ESBL	*bla*_CTX-M-1_ group, *bla*_TEM_	A	AMP, FAZ, FOV, POD, LEX, DOX, TET
77/1	Male	13 years	Owned	Unhealthy	No	Yes (cephalosporin)	ESBL	*bla*_CTX-M-1_ group, *bla*_TEM_	B2	AMP, FAZ, FOV, POD, LEX, DOX, TET, SXT
137/1	Male	1 year	Owned	Unhealthy	Yes	Yes (cephalosporin)	ESBL	*bla*_CTX-M-1_ group, *bla*_TEM_**,** *bla*_SHV_	B2	AMP, FAZ, FOV, POD, LEX, DOX, TET, SXT
161/1	Male	7 years	Owned	Unhealthy	Yes	Yes (amoxicillin + clavulanic acid)	ESBL	*bla*_CTX-M-1_ group, *bla*_TEM_	B2	AMP, FAZ, FOV, POD, LEX, AUG2, DOX, CHL, GEN, TET, SXT
195/1	Male	10 months	Owned	Unhealthy	No	Yes (metronidazole and spiramycin)	Negative	*bla*_CTX-M-1_ group, *bla*_TEM_	F	AMP, FAZ, FOV, POD, LEX, DOX, TAZ, AUG2, P/T4, IMI, GEN, ENRO, MAR, ORB, PRA

AMP: ampicillin; AUG2: amoxicillin/clavulanic acid 2:1 (value refers to amoxicillin concentration); CHL: chloramphenicol; DOX: doxycycline; ENRO: enrofloxacin; FAZ: cefazolin; FOV: cefovecin; GEN: gentamicin; IMI: imipenem; LEX: cephalexin; MAR: marbofloxacin; ORB: orbifloxacin; PRA: pradofloxacin; POD: cefpodoxime; P/T4: piperacillin/tazobactam constant 4; SXT: trimethoprim/sulfamethoxazole 1:19 (value refers to trimethoprim concentration); TAZ: ceftazidime; TET: tetracycline.

**Table 3 antibiotics-12-01249-t003:** Distribution of MICs among ESBL/AmpC-producing *E. coli* isolates.

Antimicrobial Class	Antimicrobial Agent	No. Resistant	No. of Isolates at the Indicated MIC µg/mL										
			0.12	0.25	0.5	1	2	4	8	16	32	64	128
β-Lactam (penicillins)	AMP	6/6								6			
β-Lactam (cephalosporin I)	FAZ	6/6										6	
	LEX	6/6									6		
β-Lactam (cephalosporin III)	FOV	6/6								6			
	POD	6/6								6			
	TAZ	1/6						2	3		1		
β-Lactams and β-lactamase inhibitors	AUG2	3/6						1	2	3			
	P/T4	1/6							4	1			1
Aminoglycosides	AMI	0/6						5	1				
	GEN	2/6		2	1		1			2			
Phenicol	CHL	1/6					1	3	1			1	
Fluoroquinolones	ENRO	1/6	3	2					1				
	MAR	1/6	3	2					1				
	ORB	1/6				3	2			1			
	PRA	1/6		5				1					
Tetracyclines	TET	5/6						1			5		
	DOX	6/6				2	1		1	2			
Carbapenems	IMI	1/6				5				1			
Folate pathway antagonists	SXT	3/6			3				3				

White fields denote the range of dilutions tested for each antimicrobial agent. Grey fields denote the range of dilutions not tested for each antimicrobial agent Vertical lines indicate CLSI VET01S ED6:2023 cut-off values (intermediate results were considered susceptible). AMP: ampicillin; FAZ: cefazolin; LEX: cephalexin; FOV: cefovecin; POD: cefpodoxime; TAZ: ceftazidime; AUG2: amoxicillin/clavulanic acid 2:1 (value refers to amoxicillin concentration); P/T4: piperacillin/tazobactam constant 4; AMI: amikacin; GEN: gentamicin; CHL: chloramphenicol; ENRO: enrofloxacin; MAR: marbofloxacin; ORB: orbifloxacin; PRA: pradofloxacin; TET: tetracycline; DOX: doxycycline; IMI: imipenem; SXT: trimethoprim/sulfamethoxazole 1:19 (value refers to trimethoprim concentration).

## Data Availability

The data that supported the findings of this study are available from the corresponding author upon reasonable request.

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
