# Peer review of "Fecal Carriage of Extended-Spectrum β-Lactamase-/AmpC-Producing Escherichia coli in Pet and Stray Cats"

_antibiotics, 2023, doi:10.3390/antibiotics12081249_

Round 1

Reviewer 1 Report

Comments for authors

 The manuscript entitled “Fecal carriage of extended-spectrum β-lactamase-/AmpC-producing Escherichia coli in pet and stray cats” described the isolation of E. coli from feces of owned and stray cats admitted to the Veterinary Teaching Hospital at the University of Milan, Italy during 2020 to 2022. E. coli isolates were used for determining the antibiotic resistance focused on ESBL/AmpC genotypic and phenotypic characteristics. AST was also confirmed.

 The MS has general information in the field of antimicrobial resistance. The MS has scientific soundness and results are clear. However, the MS is a preliminary survey and the sample size in this study is not sufficient for population estimation. I hope the authors will focus on the future works.

 Here are my suggestions:

 1.       Lines 117-121 should be relocated near Table 1.

2.       Line 257: the type, model, and manufacturer’s MALDI-TOF MS must be provided.

3.       Lines 249-255: Could you clarify the references for the E. coli isolation protocol? Is this standard protocol or in-house? Please revise.

4.       What is the reason why owned cats carry the ESBL/AmpC- producing E. coli more than stray cats? In my opinion, this group of cats is rising in the enclosed environment, food and drinking water might be clean, and well taken care of by their owners but they are unhealthy. This point should be added to the Discussion section.

 I have no further comments.

 Good Luck!!!!

 Kindly check the grammar, vocabulary, commas, etc.

Author Response

Dear reviewer, thank you for your precious work. Answers and comments are present in the attached file

Reviewer 2 Report

The manuscript describes ESBL/AmpC E. coli occurrence in pet and stray cats, considering potential risk factors for ESBL/AmpC phenotype development (clincial status, hospitalisation, age, etc). 

The manuscript is very interesting, although the number of cat anlysed is moderate.

The introduction supplies a concise overwiev of ESBL/AmpC E. coli diffusion in pets. However, it doesn't underline why ESBL/AmpC  phenotypes are concerning. I think it could be usefull to add a short paragraph, highlighting the importance of ESBL/AmpC strains, usually associated with increasing treatment failure, hospitalisations and mortality in humans but also in animals. I would emphasize the importance of cats as potential ESBL/AmpC reservoir, because of their wide diffusion as pet animals (already told in the introduction), the close contact and sharing common environments with human.

Material and methods:

I will remove the description of risk factors (age, ownership, etc) associated with ESBL/AmpC phenotype development from "4.1. Sample collection" paragraph and describe them in a separate paragraph. 

Results:

lines 78-81: the paragraph describes ESBL/AmpC associated risk factors of the entire collection (97 cats). However, the discussion focused only on cats associated with ESBL/AmpC E. coli carriage. This paragraph seems pointless, considering the aim of the manuscript ("estimate the presence of ESBL/AmpC producing E. coli, characterise antimcrobial resistance phenotypes and genotypes and identify risk factors associate with fecal carriage of ESBL/AmpC producing E. coli). I suggest to describe these data in the "Supplementary matherial" paragraph or to discuss them in the  "Discussion" paragraph.

line 102-103: "Resistance to.....was less frequent". I suggest to add resistance percentages in brackets.

Author Response

(The authors gave the same response as above.)

Reviewer 3 Report

Review report of Journal antibiotics(MDPI)

 Title: “Fecal carriage of extended-spectrum β-lactamase-/AmpC-producing Escherichia coli in pet and stray cats”

The study holds significant importance in the context of the global rise in antibiotic resistance. Several minor changes have been suggested and incorporated in the article as track changes. The paper may be accepted after addressing the minor revision mentioned in attached file.

Note: Please refer to the attached file for specific comments and suggested improvements.

Author Response

(The authors gave the same response as above.)

Round 2

Reviewer 2 Report

Nothing to declare